# Does Hemoperfusion Increase Survival in Acute Paraquat Poisoning? A Retrospective Multicenter Study

**DOI:** 10.3390/toxics8040084

**Published:** 2020-10-10

**Authors:** Ying-Tse Yeh, Chun-Kuei Chen, Chih-Chuan Lin, Chia-Ming Chang, Kai-Ping Lan, Chorng-Kuang How, Hung-Tsang Yen, Yen-Chia Chen

**Affiliations:** 1Department of Emergency Medicine, Taipei Veterans General Hospital Yu Li Branch, Hualien 981, Taiwan; yingtseyeh@gmail.com; 2Department of Emergency Medicine, School of Medicine, National Yang-Ming University, Taipei 112, Taiwan; cmchang5@vghtpe.gov.tw (C.-M.C.); ckhow@vghtpe.gov.tw (C.-K.H.); hjyen@vghtpe.gov.tw (H.-T.Y.); 3Department of Emergency Medicine, Chang Gung Memorial Hospital, Linkou branch, Taoyuan 333, Taiwan; master198012@gmail.com (C.-K.C.); bearuncle@yahoo.com (C.-C.L.); 4College of Medicine, Chang Gung University, Taoyuan 333, Taiwan; 5Department of Emergency Medicine, Taipei Veterans General Hospital, Taipei 112, Taiwan; kplan@vghtpe.gov.tw; 6Department of Nursing, Taipei Veterans General Hospital, Taipei 112, Taiwan; 7National Defense Medical Center, Emergency Department, Taipei 114, Taiwan

**Keywords:** paraquat, intoxication, hemoperfusion

## Abstract

The efficacy of hemoperfusion (HP) in patients with acute paraquat poisoning (PQ) remains controversial. We conducted a multi-center retrospective study to include acute PQ-poisoned patients admitted to two tertiary medical centers between 2005 and 2015. We used the Severity Index of Paraquat Poisoning (SIPP) to stratify the severity of PQ-poisoned patients. The indication to start HP was a positive result for the semiquantitative urine PQ test and presentation to the hospital was within 24 h. Early HP was defined as the first session of HP performed within five hours of PQ ingestion. A total of 213 patients (100 HP group, 113 non-HP group) were eligible for the study. The overall 60-day mortality of poisoned patients was 75.6% (161/213). Multivariate Cox regression analysis showed no statistically significant difference in 60-day survival between HP and non-HP groups (95% confidence interval (CI): 0.84–1.63, *p* = 0.363). Further subgroup analysis in the HP group showed early HP (95% CI: 0.54–1.69, *p* = 0.880), and multiple secessions of HP (95% CI: 0.56–1.07, *p* = 0.124) were not significantly related to better survival. Among acute PQ-poisoned patients, this study found that HP was not associated with increased 60-day survival. Furthermore, neither early HP nor multiple secessions of HP were associated with survival.

## 1. Introduction

Paraquat (PQ; 1,1′-Dimethyl-4,4′-bipyridinium dichloride) is a non-selective herbicide, used primarily for weed and grass control since 1960 [1]. Because PQ is highly poisonous, PQ poisoning is associated with a high mortality rate (50–90%) and is a serious public health problem, especially in Asian countries [2,3]. Worldwide, paraquat accounts for 20 deaths per million persons [4]. In the United States, the US Environmental Protection Agency classifies PQ as “restricted use.” However, the European Union and Korea have banned the use of PQ [4,5].

The initial therapy for PQ poisoning consists of gastrointestinal decontamination to prevent further absorption for oral exposures by using activated charcoal. This is followed by supportive care measures including enhanced elimination of serum PQ by hemoperfusion (HP) and medications inhibiting the inflammatory response, such as immunosuppressants (corticosteroid and cyclophosphamide) and antioxidants (N-acetylcysteine and vitamin C) [1]. HP is the most widely investigated elimination intervention because of the high clearance rate of PQ from blood [6]. However, some studies have shown that the survival benefit of HP was not as promising as expected [7,8,9].

Recently, several cohort studies have suggested a survival benefit when comparing HP to medications alone in PQ-poisoned patients [10,11,12,13]. Many of these studies demonstrated that early initiation (within 4–6 h) of HP after acute PQ ingestion might improve the survival of PQ-poisoned patients [10,12,13]. However, these studies may have limitations as the authors did not clearly stratify patients by poisoning severity or account for other interventions [12,13]. For example, some studies did not report the elapsed time to the first HP, and these studies did not include immunosuppressants or antioxidants as part of the therapy [11]. Additionally, these studies were conducted in developing countries such as China or India, where care may be delayed, especially in rural areas [11,13].

We conducted a multi-center retrospective study to improve the characterization of the clinical efficacy of HP in PQ-poisoned patients. We stratified patients using the Severity Index of Paraquat Poisoning (SIPP) and measured the use of immunosuppressive and antioxidant treatments. SIPP is calculated by multiplying the elapsed time (hours) from ingestion to arrival by the serum paraquat level (mcg/mL). Sawada et al. reported that patients with a SIPP of less than 10 have a good chance of survival. In contrast, those with a SIPP between 10 and 50 often die because of interstitial pulmonary fibrosis secondary to PQ poisoning, and those with values greater than 50 die rapidly owing to circulatory collapse [14]. We also evaluated whether early HP and multiple sessions of HP were associated with improved survival.

## 2. Materials and Methods

### 2.1. Study Population

This study was a multi-center retrospective chart review study, performed in two tertiary referral medical centers in northern Taiwan: Taipei Veterans General Hospital and Linkou Chang Gung Memorial Hospital. We collected cases of acute PQ poisoning admitted to the emergency department from 1 January 2005, to 31 December 2015. We performed patient screening by collecting those whose emergency department (ED) discharge diagnosis was pesticide intoxication from both hospitals. We further confirmed the case to be acute PQ poisoning by directly reviewing the electronic medical records. The inclusion criteria were: 1. confirmed acute paraquat poisoning cases; and 2. age greater than 16 years old. The exclusion criteria were: 1. the time from ingestion to arrival at ED longer than 24 h; 2. missing data on serum PQ concentrations; 3. missing of mortality status or interval to death; and 4. non-oral exposure.

### 2.2. Ethics Statement

This study complied with the guidelines of the Declaration of Helsinki, and the Institutional Review Board (IRB) also approved it (IRB approval codes and dates: 2016-05-001BC on 13 May 2016; 201600407B0 on 21 April 2016). All the data were analyzed anonymously, and the IRB waived the requirement for informed consent.

### 2.3. Data Collection and Definition

Two board-certified emergency physicians independently performed data extraction using a standardized procedure in each case. The kappa coefficient was 0.780. We recorded patients’ baseline characteristics (sex, age), hospitals, exposure history (amount, reason), initial vital signs and routine labs on arrival, initial serum PQ concentration (mcg/mL), and interval from ingestion to the medical facility (hour, time to physician provider). SIPP was calculated in each case. The treatment modalities identified for each patient included gastric decontamination with activated charcoal, HP, immunosuppressive therapy, and antioxidant therapy. For HP, we recorded the interval from ingestion to initiation of HP and the secession of HP performed. Each secession of HP was performed in a six hour duration. The indication to start HP was a positive result for semiquantitative urine PQ test and presentation to the hospital within 24 h. The early HP was defined as PQ-poisoned patients who received the first session of HP within five hours of PQ ingestion.

The protocol for immunosuppressive therapy consisted of administration of corticosteroid, starting with methylprednisolone (1 g/day for three days) followed by dexamethasone (20–40 mg per day) or administration of cyclophosphamide, starting dose with 15 mg/kg/day for two days. The protocol for antioxidant therapy consisted of administration of N-acetylcysteine with a loading dose 140 mg/kg, then 70 mg/kg Q4H for three days, and vitamin C (200–400 mg per day).

### 2.4. Outcome Measures

The primary outcome measures were 60-day survival and time to death. Subgroup analysis included 60-day survival in PQ-poisoned patients with early HP and other prognostic factors.

### 2.5. Statistical Analysis

We performed all statistical analyses with IBM^®^ SPSS version 25 (IBM Corp, Armonk, NY, USA). Independent *t*-tests and Chi-squares test were used for baseline data between HP and non-HP groups. Continuous variables are expressed as the median and standard deviation (SD), and categorical variables are expressed as numbers or percentages for each item. To assess the relationship between treatment protocols and mortality, the Kaplan–Meier survival curves were compared with the log rank test. Then we performed a Cox proportional hazard regression model to determine the primary outcomes while controlling for possible confounding baseline characteristics in all patients, including age, sex, SIPP, and initial serum concentration of creatinine. To evaluate the benefit of early initiation of HP within five hours, we performed another Cox proportional hazard regression model among patients who underwent HP. For every Cox model, we reported the hazard ratios (HR) for survival, and 95% confidence intervals (CIs). All probabilities were two-sided, with *p* < 0.05 considered statistically significant.

## 3. Results

### 3.1. Enrollment

Of all the pesticide intoxication cases, including paraquat poisoning, organophosphate poisoning, carbamate poisoning, glyphosate poisoning, etc., we identified 320 patients (34%) with acute PQ poisoning during the studying period. Of these, 107 patients were excluded. Finally, we included 213 patients in the study for analysis (Figure 1).

To evaluate for selection bias, we compared the baseline characteristics between the inclusion group (*n* = 213) to those who presented early (≤24 h), were orally exposed, and were missing serum PQ concentration (*n* = 24). There was no significant difference in terms of mortality between the two groups (Table 1).

### 3.2. Baseline Characteristic

The mean age was 51.4 ± 17.4 years, with age ranging between 17 and 90 years. Most of these patients were healthy at baseline. The overall mortality was 75.6% (161/213). Of the 161 expired patients, 129 (80.1%) patients died within three days after PQ ingestion. The mean time to mortality in HP group and non-HP group were 4.2 days and 2.8 days (*t*-test *p* = 0.158). The mortalities for patients who had SIPP < 10, SIPP 10–50, and SIPP > 50 were 33.8% (25/74), 96.9% (63/65), and 98.6% (73/74), respectively. The mean serum paraquat level was 34.24 mcg/mL. The mean elapsed time from ingestion to ED arrival was 5.91 h; only 32.4% (69/213) presented within three hours. One hundred and fifty (70.4%) patients received gastric decontamination. One hundred (46.9%) patients received HP, and twenty-two of them got early HP. The percentage of the patients treated with immunosuppression therapy and antioxidant therapy was 68.5% (*n* = 146) and 40.0% (*n* = 85).

### 3.3. Differences between HP and Non-HP Groups

The analysis of characteristics revealed younger age, higher Glasgow Coma Scale (GCS) score on arrival, and lower creatinine concentration in the HP group. Additionally, the HP group had a significantly higher proportion administered cyclophosphamide and lower proportion administered N-acetylcysteine. The mortality for the HP group and the non-HP group did not show a statistically significant difference (Table 2).

### 3.4. Survival Analysis

Our multivariate Cox analysis found no statistically significant difference in survival between HP and non-HP groups (HR: 1.17, 95% CI: 0.84–1.63, *p* = 0.363) after adjusting to confounding factors (Table 3). Neither did Kaplan–Meier survival analysis show improved survival in HP group (Appendix A). Younger age, lower initial creatinine level, and lower SIPP scores were significantly related to better survival (Table 3).

In the subgroup analysis of the 100 patients who received HP, the survival time of the early HP group was not significantly higher than the late HP group (elapsed time > 5 h; Figure 2).

In multivariate Cox analysis, both early HP and multiple secessions of HP were not significantly related to better survival. A lower SIPP score was the only variable significantly associated with better survival (HR: 2.49, 95% CI: 1.75–3.56, *p* < 0.001; Table 4).

## 4. Discussion

This study found that HP was not associated with increased survival of acute PQ-poisoned patients after adjustment for SIPP, age, and renal function. Furthermore, neither early HP nor multiple secessions of HP were associated with survival. To our knowledge, this study was the largest cohort to report the clinical impact of HP among PQ-poisoned patients in Taiwan.

Hemoperfusion is frequently recommended for acute PQ poisoning. Previous studies have suggested HP is associated with better survival when compared to medications alone in PQ-poisoned patients [10,11,12,13]. However, these results might have been biased by lack of quantitative serum PQ concentration that led to misclassification in survival analysis [10,12,13] or because they did not include the use immunosuppressant or antioxidant therapies [11,12,13]. PQ concentrations in urine and serum have been used to predict the severity of poisoning [1]. However, blood concentrations are a better predictor of outcome [15]. The SIPP, an index for clinicians to determine the severity of PQ poisoning, is determined by multiplying serum PQ concentration on admission (ppm) by the time to treatment (in hours). SIPP values under 10 are an indication to clinicians that PQ-poisoned patients have a higher probability of survival [14]. In addition, when compared to other scoring systems such as APACHE II (Acute Physiology and Chronic Health Evaluation score), SIPP has been proven to be superior in predicting the prognosis [16]. After stratifying patients using SIPP, our study found that HP was not associated with a survival benefit of PQ-poisoned patients. These results were similar to those studies using the stratification of PQ concentration in serum [7,8,9].

Early initiation of HP (compared to later initiation) was also investigated recently [10,12,13]. It is based on the idea that the peak time of plasma PQ absorption is only 1–3 h [6]. Once the plasma PQ concentration reaches the peak level, PQ rapidly distributes into peripheral tissues and lungs. Some cohort studies found that early initiation of HP (within 4–6 h) after acute PQ ingestion might improve the survival of PQ-poisoned patients [10,12,13]. However, our result suggests no survival benefit when HP was initiated within 5 h. There are several possible reasons for these differences. First, treatment within the first five hours has a minimal impact on PQ absorption into the lung. The concentration peak time of PQ in the lung is approximately 5–7 h [17]. Early HP therapy does not reduce the concentration of PQ in the muscle and lung tissues [7]. Second, PQ-poisoning related multi-organ failure and death include the generation of free radicals, the subsequent transcription of pro-inflammatory proteins, and direct damage to mitochondria [18]. There is no evidence that HP has clearance on free radicals or cytokines in human species. Third, in reality, the initiation of HP within five hours was very impractical since most patients presented to the hospital for more than 5 h after exposure. In our study, only 10.3% (22/213) patients received HP within five hours from PQ ingestion. The mortality of this proportion of patients was 77% (17/22), which was higher than retrospective studies conducted in Taiwan and China [10,12].

The HP group was significantly younger, had better GCS, better renal function, and lower SIPP scores than the non-HP group (Table 2). This result revealed that many physicians tend to initiate HP therapy to younger patients or those presented without signs of end-organ dysfunction, such as good GCS and renal function. They believe that these patients would gain some benefit from enhanced PQ clearance. On the contrary, if patients presented with multi-organ dysfunction on arrival, physicians may be reluctant to initiate HP therapy. In this kind of condition, HP has little effect in clearing the free radical and cytokines already generated from paraquat intoxication unless the polymyxin B immobilized fiber column was used [19]. In this cohort study, however, we found the HP group was not significantly related to increasing survival (*p* = 0.363, adjusted hazard ratio (HR): 1.17); in Cox regression model even these patients were younger or had better physiological status than non-HP group. This result indicated that HP therapy might not increase the survival of acute paraquat poisoning.

Multiple secessions of HP have also been suggested as a reasonable method to remove more PQ because there is often a rebound in plasma PQ concentrations following the termination of the extracorporeal procedure [20]. Prior studies have also noted that multiple secessions of HP were associated with improved survival [21,22]. However, our result does not show a difference between a single and multiple sessions of HPs in terms of survival. One potential explanation is that PQ kinetics are described by the two-compartment model. As muscle is the primary source of PQ reservoir after initial distribution, the muscle slowly releases PQ back into the circulation over days to weeks with a half-life ranging from 100–150 h [23]. In addition, PQ mainly accumulates in the lung (pulmonary concentrations can be six to ten times higher than those in the plasma), where it is retained even when blood concentrations start to decline [1]. Therefore, total body PQ burden was not significantly reduced when multiple secessions of HP were performed [24]. Another potential explanation is that most of the PQ-poisoned patients (80.1%) in our study died within three days, so we did not have enough patients to measure the effect of multiple secession of HP on pulmonary fibrosis.

Recently several studies conducted in China reported that HP combined with continuous renal replacement therapy (CCRT) might improve the survival of mildly to moderately PQ intoxicated patients [25,26]. These studies reported that CCRT could reduce the harm of PQ redistribution from peripheral tissue. The mortality in these studies was around 30–40%. However, the mortality in non-HP group who had a SIPP < 10 was 25.6% (10/39), which was similar to previous studies where HP was not administered. PQ is primarily excreted in urine [1], and the renal clearance rate for PQ under normal renal function is two to four times the clearance rate of HP [17]. Therefore, the clearance rate of PQ by HP is minimal when the plasma PQ level is low [27]. In our study, a low serum creatinine was associated with a good prognosis of PQ-poisoned patients. It is doubtful that HP can provide additional PQ clearance in mildly PQ-poisoned patients with normal renal function. Therefore, more studies are needed to investigate if CCRT confers an additional survival benefit.

In conclusion, we demonstrated that single or multiple secessions of HP with an elapsed time from exposure to first HP within five hours did not improve survival in PQ-poisoned patients. Nevertheless, we found lower SIPP values, lower serum creatinine concentration, and younger age were prognostic factors for all PQ-poisoned patients. The strength of our study is risk stratification for the severity of PQ poisoning and delay to care, and the use of immunosuppressants/antioxidants.

## 5. Limitation

Our study has certain limitations. First, this is not a prospective controlled trial. Patients were not randomized to treatment, therefore selection bias is possible. While we adjusted for some potential confounders, we cannot adjust for all baseline characteristics. Second, the number of cases was small, and the study population comprised of only Asians. Third, we lack the serial serum PQ level, which makes it difficult to evaluate the actual toxicokinetic effect from HP.

## Figures and Tables

**Figure 1 toxics-08-00084-f001:**
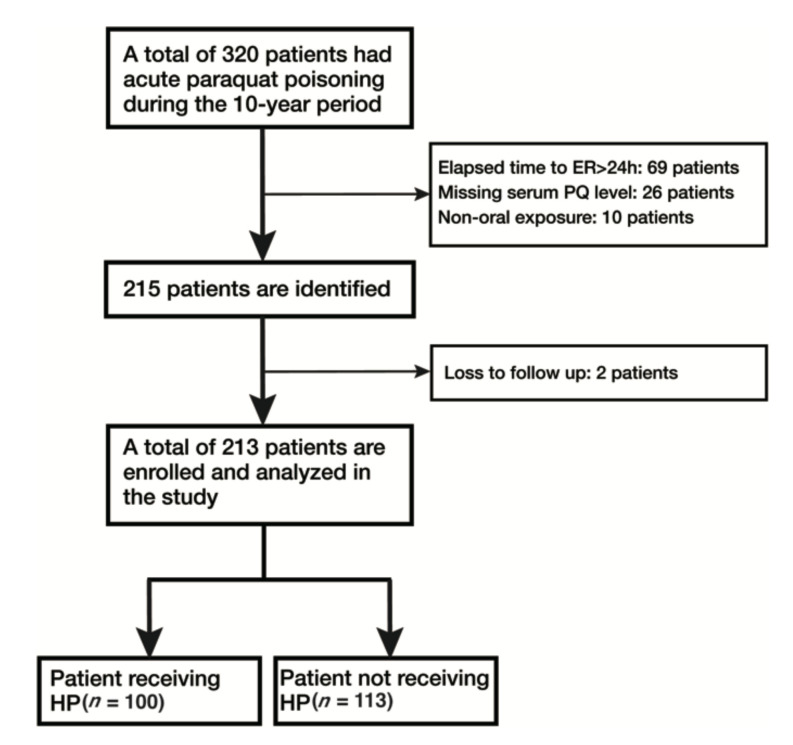
Inclusion algorithm.ER: Emergency room; PQ: paraquat; HP: Hemoperfusion.

**Figure 2 toxics-08-00084-f002:**
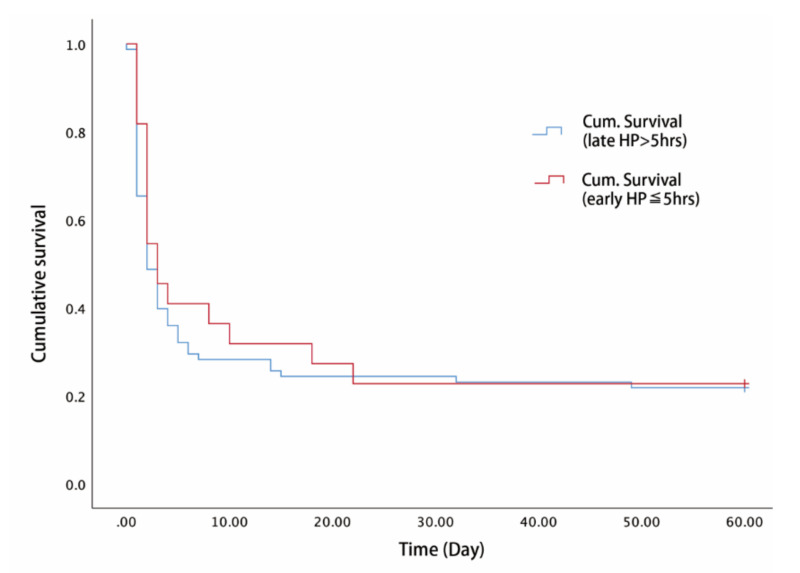
Kaplan–Meier survival analysis of between the early HP group and late HP group (*p* = 0.603, log-rank test).

**Table 1 toxics-08-00084-t001:** Baseline characteristics comparison for the evaluation of selection bias.

Parameter	Patient Having sPQ, *n* = 213	Patient Not Having sPQ, *n* = 24	*p*-Value
Median age, years (SD)	51 (17.44)	54 (17.84)	0.391
Gender			0.488
Male, *n* (%)	145 (68.1)	18 (75.0)	
Female, *n* (%)	68 (31.9)	6 (25.0)	
GCS > 13, *n* (%)	154 (72.3)	14 (58.3)	0.153
Creatinine (mg/dl); Mean (SD)	1.66 (1.29)	2.68 (4.00)	0.227
Gastric lavage, *n* (%)	171 (80.3)	17 (70.3)	0.291
Activated charcoal, *n* (%)	150 (70.4)	11 (45.8)	0.014
Cyclophosphamide, *n* (%)	32 (15.0)	3 (12.5)	1.0
N-acetylcysteine, *n* (%)	74 (34.7)	10 (41.7)	0.501
Vitamin C, *n* (%)	48 (22.5)	6 (25.0)	0.785
Steroid, *n* (%)	146 (68.5)	18 (75)	0.516
HP, *n* (%)	100 (46.9)	5 (20.8)	0.015
Mortality, *n* (%)	161 (75.6)	22 (91.7)	0.075

sPQ: serum paraquat level; GCS: Glasgow Coma Scale; SD: Standard deviation; HP: Hemoperfusion. *p* > 0.05 non-significant.

**Table 2 toxics-08-00084-t002:** Baseline characteristics between the HP group and non-HP group.

Parameter	HP Group *n* = 100	Non-HP Group *n* = 113	*p*-Value
Median age, years (SD)	47 (17.41)	54 (16.96)	0.006
Gender			0.365
Male, *n* (%)	65 (65.0)	80 (70.8)	
Female, *n* (%)	35 (35.0)	33 (29.2)	
GCS > 13, *n* (%)	82 (82)	72 (63.7)	0.003
Creatinine (mg/dL); Mean (SD)	1.36 (0.82)	1.93 (1.56)	0.001
SIPP; Mean (SD)	98.71 (399.10)	158.53 (338.07)	0.238
SIPP < 10, *n* (%)	35 (35.0)	39 (34.5)	0.941
SIPP 10–50, *n* (%)	33 (33.0)	32 (28.3)	0.459
SIPP > 50, *n* (%)	32 (32.0)	42 (37.2)	0.429
Gastric lavage, *n* (%)	83 (83.0)	88 (77.9)	0.348
Activated charcoal, *n* (%)	75 (75.0)	75 (66.4)	0.169
Cyclophosphamide, *n* (%)	23 (23.0)	9 (8.0)	0.002
N-acetylcysteine, *n* (%)	22 (22.0)	52 (46.0)	<0.001
Vitamin C, *n* (%)	21 (21.0)	27 (23.9)	0.614
Steroid, *n* (%)	73 (73.0)	73 (64.6)	0.188
Mortality, *n* (%)	78 (78.0)	83 (73.5)	0.441

GCS: Glasgow Coma Scale; SIPP: Severity Index of Paraquat Poisoning; *p* > 0.05 non-significant.

**Table 3 toxics-08-00084-t003:** Multivariate Cox Proportional Hazards Model for Mortality Prediction of Paraquat Poisoning in all patients (*n* = 213).

Variables	Adjusted HR (95% CI)	*p*-Value
Age (years)	1.01 (1.00–1.02)	0.024
Gender as female (Yes)	0.97 (0.69–1.37)	0.888
Creatinine (mg/dL)	1.17 (1.05–1.29)	0.003
SIPP	2.61 (2.08–3.28)	<0.001
Cyclophosphamide (Yes)	0.80 (0.50–1.33)	0.384
N-acetylcysteine (Yes)	1.00 (0.71–1.41)	0.995
HP group (Yes)	1.17 (0.84–1.63)	0.363

HR: Hazard ratio; SIPP: Severity Index of Paraquat Poisoning; HP group: Hemoperfusion group; *p* > 0.05 non-significant.

**Table 4 toxics-08-00084-t004:** Multivariate Cox Proportional Hazards Model for Mortality Prediction of Paraquat Poisoning in HP group (*n* = 100).

Variables	Adjusted HR (95% CI)	*p*-Value
Age (years)	1.01 (0.99–1.03)	0.083
Gender as female (Yes)	0.84 (0.50–1.39)	0.490
Creatinine (mg/dL)	1.15 (0.88–1.50)	0.320
SIPP	2.49 (1.75–3.56)	<0.001
Cyclophosphamide (Yes)	0.71 (0.39–1.31)	0.277
N-acetylcysteine (Yes)	0.68 (0.36–1.27)	0.224
Secession of HP	0.78 (0.56–1.07)	0.124
Interval to first HP ≤ 5 h	0.96 (0.54–1.69)	0.880

SIPP: Severity Index of Paraquat Poisoning; HP: Hemoperfusion; *p* > 0.05 non-significant.

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
