# Peer review of "Does Hemoperfusion Increase Survival in Acute Paraquat Poisoning? A Retrospective Multicenter Study"

_toxics, 2020, doi:10.3390/toxics8040084_

Round 1

Reviewer 1 Report

The authors conducted a multi-center retrospective study to investigate the efficacy of hemoperfusion (HP) in patients with acute paraquat poisoning (PQ). This study showed HP was not associated with increased 60-day survival. Furthermore, neither early HP nor multiple secessions of HP were
associated with survival. This study is very informative and very useful and therefore I recommend to accept in the present form. 

Reviewer 2 Report

The authors provide a valuable contribution to the discussion about the efficacy of hemoperfusion in the treatment of acute paraquat poisoning. The study is well conceptualized and statistical evaluation is sound. English writing is good, but needs some improvements, mainly in terms of wording and style. There are a few minor points that should be addressed before a publication can be recommended.

1. The authors provide the mean SIPP, but not the mean serum concentrations and the mean time to treatment. However, blood levels and their variance are of high interest for comparability. For example, it appears at least possible that a significant number of patients were poisoned by suicidal attempts. In this case, rather high serum levels can be imagined, while lower values with larger variance may occur in accidental intake. It is currently not clear if the presented scenario and consequently the determined efficacy of hemoperfusion, is representative or if the serum levels were rather high in this study.

2. Relevant information may be found in the comparison of time to death between the treated and the untreated group within the 60-day observation period. The authors state that 80.1% of patients passed away within three days of PQ ingestion. However, is there a difference in mean time to death between the HP- and non-HP group? In survival analysis, outcomes that are not solely event based should be included if possible.

Wording:

line 18: Sentence incomplete:..HP was a positive....

line 21: The overall 60-day mortality of poisoned patients was....

line 33: PQ poisoning is associated with a high mortality rate.

line 35: ... the European Union and Korea...

line 36: ...gastrointestinal decontamination...

line 52: ...study to improve the characterization of the clinical efficacy of HP...

line 62: Hospital specific number codes are not of relevance for the general audience.

line 65: While the meaning is clear, "both male and female" cannot be a criteria.

line 66 : ...missing data on PQ serum concentrations...

line 74: "Any dispute regarding data retrieval...": Please explain or omit sentence.

line 118: "To evaluate...": Sentence needs to be rephrased.

Figure 2: Legend shows 2x early HP.

line 174: ...time to treatment (in hours)

line 180: "Early initiation of HP..." or "Early onset of HP..."

line 185: HD: Please explain abbreviation at first use in text.

line 186: "The 5 hours window" Better: Treatment within the first five hours has....

Reconsider when using numerals or written numbers in text. Typically, numbers below10 are shown as words and larger numbers as digits....

line 194: "The result was higher than..": "The proportion of patients receiving HP within five hours was higher..."

line 198-204: This section is not totally clear. Please rephrase....

line 212: "...our result does not show a difference between a single HP and multiple repeated HPs in terms of survival."

line 214: releases

Reviewer 3 Report

In the present manuscript, Yeh et al report a statistic retrospective study to evaluate the efficacy of hemoperfusion (HP) on acute paraquat (PQ) poisoning. Paraquat is a highly toxic herbicide mainly used in Asian countries, thus paraquat poisoning remains a serious public health concern in these countries. In the current manuscript, although HP is widely employed for the removal of serum paraquat, authors showed that there is no significant difference on 60-day survival between HP and non-HP group. In addition, either early HP or multisession of HP had no remarkable impact on the survival of PQ-poisoned patients.

The strategy, objectives and statistic results are clearly described and could be useful when considering the use of HP as the therapy for PQ poisoning. On the other hand, additional information and minor correction would be necessary before considering the publication as described below.

1) In introduction, it would be helpful to add more general information such as definition of the SIPP (Severity Index of Paraquat Poisoning), number of paraquat poisoning and death reported per year if available. 

2) Authors are focusing on only one factor (SIPP) to evaluate the efficacy of hemoperfusion, however, practically many hospitals are not always equipped with the facilities to measure the serum paraquat levels. Why didn't authors perform statistic analyses using other factors such as APACHE II, MSAPS II and MSAPS IIe to evaluate the efficacy of HP? It would also give us even more comprehensive information if authors perform the comparison between these factors.

3) As minor correction, Line 31, introduction, the name of Paraquat must be corrected as 1,1'-Dimethyl-4,4'-bipyridinium dichloride. 
